# Advancing Extrapulmonary Tuberculosis Diagnosis: Potential of MPT64 Immunochemistry-Based Antigen Detection Test in a High-TB, Low-HIV Endemic Setting

**DOI:** 10.3390/pathogens14080741

**Published:** 2025-07-28

**Authors:** Ahmad Wali, Nauman Safdar, Atiqa Ambreen, Asif Loya, Tehmina Mustafa

**Affiliations:** 1Centre for International Health, Department of Global Public Health and Primary Care, University of Bergen, Arstadveien 21, 5009 Bergen, Norway; tehmina.mustafa@uib.no; 2Department of Health, Government of Balochistan, Civil Secretariat, Zarghoon Road, Quetta 87300, Pakistan; 3Interactive Research and Development, Level 10, One George Street, Singapore 049145, Singapore; safdar.nauman@gmail.com; 4Department of Microbiology, Gulab Devi Teaching Hospital, Lahore 05441, Pakistan; atiqaambren@gmail.com; 5Department of Pathology, Shaukat Khanum Memorial Cancer Hospital and Research Center, 7A Block R-3 MA Johar Town, Lahore 54000, Pakistan; asifloya@skm.org.pk; 6Department of Thoracic Medicine, Haukeland University Hospital, Bergen, Haukelandsveien 22, 5009 Bergen, Norway

**Keywords:** pleuritis, lymphadenitis, immunocytochemistry, immunohistochemistry, Pakistan

## Abstract

Extrapulmonary tuberculosis (EPTB) remains diagnostically challenging due to its paucibacillary nature and variable presentation. Xpert and culture are limited in EPTB diagnosis due to sampling challenges, low sensitivity, and long turnaround times. This study evaluated the performance of the MPT64 antigen detection test for diagnosing EPTB, particularly tuberculous lymphadenitis (TBLN) and tuberculous pleuritis (TBP), in a high-TB, low-HIV setting. Conducted at Gulab-Devi Hospital, Lahore, Pakistan, this study evaluated the MPT64 test’s performance against conventional diagnostic methods, including culture, histopathology, and the Xpert MTB/RIF assay. Lymph node biopsies were collected, and cell blocks were made from aspirated pleural fluid from patients clinically presumed to have EPTB. Of 338 patients, 318 (94%) were diagnosed with EPTB. For TBLN, MPT64 demonstrated higher sensitivity (84%) than Xpert (48%); for TBP, the sensitivity was 51% versus 7%, respectively. Among histopathology-confirmed TBLN cases, MPT64 outperformed both culture and Xpert (85% vs. 58% and 47%). Due to the low number of non-TB cases, specificity could not be reliably assessed. The MPT64 test shows promise as a rapid, sensitive diagnostic tool for EPTB, particularly TBLN, in routine settings. While sensitivity is notably superior to Xpert, further studies are needed to evaluate its specificity and broader diagnostic utility.

## 1. Introduction

The diagnostic challenges of extrapulmonary tuberculosis (EPTB) are multifaceted and stem from a combination of biological, logistical, and infrastructural issues. Accurate diagnosis of EPTB requires specific tests to detect mycobacteria, but medical professionals face challenges due to diverse clinical manifestations, the paucibacillary nature of the disease, and the fact that its diagnosis typically requires invasive procedures to obtain tissue samples, which depend on skilled personnel and functional laboratory system resources that are often limited in low- and middle-income countries (LMICs) [1,2].

EPTB accounts for approximately 20–30% of all active tuberculosis (TB) cases globally, with this proportion rising to 50% among individuals co-infected with human immunodeficiency virus (HIV) infections [3,4,5]. Tuberculous lymphadenitis (TBLN) is the most common expression, comprising 35–50% of cases, while tuberculous pleuritis (TBP) is the second most predominant manifestation, which accounts for 10–16% [3,6,7]. Combined, these two forms represent about half (50%) of all EPTB presentations. Globally, 8–10 million people are infected with TB annually, with EPTB accounting for up to 17% of all TB cases reported in 2023 [8]. The majority (80%) of incident TB cases and mortalities occur in low- and middle-income countries. Pakistan is among the high-TB-burden countries; 428,600 TB cases were notified in 2023, with 18% (77,148) being EPTB [8,9].

Despite significant advances in the field of TB diagnosis over the last decade, including advancements in microscopy, culture, and molecular techniques, the diagnosis of EPTB remains a major challenge [10,11]. The clinical and histological diagnostic criteria utilized have low sensitivity and specificity, which might lead to missed or overdiagnosis [12]. The Xpert MTB/RIF assay (Xpert), a significant milestone in TB diagnostics, shows reduced efficacy for EPTB due to variability in specimen types, limited tissue availability, and the need for complex sample processing [13,14]. Isolation of *Mycobacterium tuberculosis* (MTB) from clinical samples by culture remains the “gold standard” due to its high specificity, ability to detect viable bacilli (10–100 bacilli/mL) from biological samples, and ability to identify species. However, low sensitivity and long turnaround times limit the use of culture in resource-limited settings [1,15]. Due to the absence of a low-cost, robust, rapid, and accurate diagnostic method, diagnosis of EPTB is often delayed or misdiagnosed, resulting in increased disease severity and mortality. Therefore, there is an urgent need for a better diagnostic test that is feasible and sustainable in resource-limited settings.

MPT64 is one of the major culture filtrate proteins (24 kDa) encoded by the RD2 region genes, secreted by MTB complex species, and is a specific antigen that differentiates it from the Mycobacterium other than tuberculosis species. The diagnostic potential of an immunochemistry-based MPT64 antigen detection test (MPT64 test) has demonstrated greater sensitivity compared with conventional methods and the Xpert. Its sensitivity and specificity are also comparable with the nested polymerase chain reaction (nested-PCR) [15,16,17,18]. Results from earlier research indicate that MPT64 has a unique capacity for intracellular accumulation that enables the detection of smaller numbers of mycobacteria in the lesion, which is not achievable using other conventional methods for the diagnosis of TB [15,19].

This study aimed to evaluate the diagnostic performance of the MPT64 test for TBLN and TBP in routine diagnostics in a high-TB, low-HIV setting. Specifically, we compared its sensitivity to that of the Xpert assay and assessed the relationship between MPT64 antigen levels and diagnostic delays, disease severity, and histopathological features among affected patients.

## 2. Materials and Methods

### 2.1. Study Setting and Population

This study was part of a larger multicenter study aimed at evaluating the new diagnostic MPT64 test for the diagnosis of EPTB [12,18,20,21]. It was conducted at the Gulab Devi Hospital (GDH), a private not-for-profit tertiary-care hospital and the Shaukat Khanum Memorial Cancer Hospital and Research Centre (SKMCH&RC), a tertiary care cancer center, in Lahore, Pakistan. The GDH was primarily responsible for patient recruitment, routine diagnostics, treatment and follow-up, and the collection of biological samples for testing. Whereas, SKMCH&RC was responsible for performing immunocytochemistry (ICC) and immunohistochemical staining (IHC) for MPT64. Patients of all ages presenting with clinical suspicion of EPTB were prospectively enrolled in the outpatient department between April 2016 and August 2017. Clinical specimens were collected during the initial presentation/enrollment period prior to the initiation of anti-tuberculosis treatment (ATT). Comprehensive imaging or microbiological evidence to confirm concurrent pulmonary involvement was not consistently available and, therefore, not analyzed. Enrollment was facilitated by a hospital physician assisted by a trained research team. Patients who had not given informed written consent, those who had received ATT in the previous year and had known HIV infection, or those under active immunosuppressive therapy were excluded.

### 2.2. Study Questionnaire and Data Collection

The questionnaire used in this study for data collection was adapted from a multicenter study [22], with slight modifications. The English version was translated into the national language of Urdu and then back-translated to English by an independent translator. Consistency was checked between the original version and the back-translated version to ensure their validity. Sociodemographic, clinical, laboratory, and radiological data were extracted from patients’ medical record files from the hospital, including culture, Xpert, acid-fast bacilli (AFB) staining, histology, cytology, ICC, IHC, and relevant radiological imaging.

### 2.3. Biological Sample Collection, Processing, and Transportation

For presumptive TBLN, excisional lymph node biopsies were obtained as per the developed standard sterile protocols of GDH. Each biopsy specimen was divided into two parts: one portion was placed in 10 mL of 0.9% physiologic saline for microbiological analysis (AFB staining, culture on solid and liquid media, and Xpert) [21], and the other portion was fixed in buffered formalin for histology and ICC/IHC staining.

Pleural fluid was aspirated aseptically from patients presumptive for TBP. Half of the sample was subjected to routine TB diagnostics (acid-fast staining, culture on solid and liquid media, and Xpert); the remaining half was processed to make cell blocks by centrifuging the specimen at 2500 rpm for 15 min within one hour of aspiration. The supernatant was discarded, and the deposit was mixed with 10% formalin (1:10 dilution) and left for 24 h for fixation. After fixation, the sample was centrifuged again, the supernatant was discarded, and the deposit was placed on a filter paper, which was placed in a cassette for processing in the histopathology laboratory. All samples intended for ICC/IHC analysis were transported to SKMCH&RC, Lahore, and the new test reagents and relevant orientation were provided to the laboratory to perform the MPT64 test.

### 2.4. Assessment of Immunostainings with MPT64

ICC and IHC for MPT64 were performed using an in-house polyclonal anti-MPT64 antibody (1:250 dilution), with detection via the Dako Envision+ System-HRP kit (K4009, Dako, Glostrup, Denmark). For ICC, smears were rehydrated, endogenous peroxidase activity was blocked, and slides were incubated with the primary antibody for 60 min, followed by HRP-conjugated polymer detection for 45 min. Visualization was achieved using a 3-amino-9-ethylcarbazole (AEC) substrate and counterstaining with Mayer’s hematoxylin. For IHC, formalin-fixed, paraffin-embedded sections underwent deparaffinization and microwave-based antigen retrieval in citrate buffer (pH 6.2) before following the same staining protocol as ICC. Slides were mounted using Immu-Mount (Thermo Fisher Scientific, Waltham, MA, USA), and wash steps were carried out using Dako Wash Buffer (S3006, Dako, Glostrup, Denmark) [15,23,24].

Slides prepared from lymph node biopsies and pleural fluid cell blocks were independently assessed by two pathologists at SKMCH&RC who were blinded to the clinical categorization and diagnostic data of the patients to eliminate bias. Slide screening was initially done at low magnification (10×), followed by a more detailed examination at higher magnification (40×). Figure 1 illustrates a positive MPT64 stain. MPT64 test staining was considered positive if granular reddish-brown deposits were observed either intracellularly in inflammatory cells or extracellularly in areas of necrosis. The final classification was based on the consistent presence of this stain pattern in the relevant tissue compartments. The intensity of staining was used as an estimation of the antigen load, from mild to moderate and severe.

### 2.5. Patient Categorization and Disease Severity Criteria

Patients were categorized using a composite reference standard (CRS). Table 1 shows the CRS used to classify the patients into four different categories by combining several diagnostic criteria that were devised based on clinical signs and symptoms, radiological findings, results from various laboratory tests, response to ATT, and response to specific non-TB therapy. Disease severity was dichotomized as less severe or severe based on symptom burden, radiographic evidence, and cytology findings, in accordance with WHO TB diagnostic and operational guidelines [25,26].

### 2.6. Study Flow

Figure 2 shows the patient inclusion criteria, flow, and biological specimen collection. During the study period, a total of 373 presumptive TBLN and TBP patients were screened for eligibility. After exclusion due to prior ATT and consent refusal, 338 patients were included in the final analysis. Among them, as per the CRS, 150 (44%) were categorized as confirmed, 137 (41%) as probable, 33 (10%) as possible EPTB cases, and 18 (5%) as non-EPTB cases.

### 2.7. Statistical Analysis

Data were double-entered and validated using EpiData software (version 3.1 for entry and 2.2.2.183 for review, EpiData Association, Odense, Denmark). Descriptive statistics (means, medians, and interquartile ranges) were calculated. Categorical variables were compared using Pearson’s chi-square test, and continuous variables using non-parametric tests (e.g., Mann–Whitney U test). To compare diagnostic accuracy, odds ratios (ORs) and 95% confidence intervals (CIs) were estimated using logistic regression. Confounding variables were adjusted for where appropriate. A *p*-value < 0.05 was considered statistically significant. No imputation was applied for missing values; only complete cases were analyzed.

### 2.8. Ethics Approval and Consent to Participate

Ethical clearance was obtained from the Regional Committee for Medical and Health Research Ethics, Western Norway (REK Vest) [project number 234457], the National Bioethics Committee of Pakistan, and the Institutional Review Board (IRB), GDH, and IRB at SKMCH&RC, Lahore, Pakistan. All study participants signed written informed consent. Parents and/or legal guardians of participants under the age of 15 signed the informed consent. All procedures were performed in accordance with relevant guidelines and regulations.

## 3. Results

### 3.1. Patient Diagnostics and Clinical Characteristics

Table 2 describes the baseline sociodemographic and clinical characteristics of the study participants. Out of 338 assessed patients, 320 (95%) had an EPTB diagnosis, while 18 (5%) had a non-EPTB diagnosis. The majority of EPTB patients (72%) were young adults, 15–44 years old, and the non-EPTB patients were also predominantly young adults. Among the EPTB patients, TBLN was observed in 186 (58%), while TBP accounted for 134 (42%). TBLN was predominant in females (117/186; 63%), whereas TBP was more common among males (86/134; 64%). HIV status was available for 80 patients; none of those tested were positive.

Table 3 presents the results of the diagnostic tests for both the pleural effusion aspirates and lymph node biopsies. Among all EPTB cases, the MPT64 test yielded positive results in 67% of cases, compared to 2% with AFB staining, 29% with Xpert, and 44% with culture. MPT64 test positivity was higher in lymph node biopsies (84%) than in pleural fluid aspirate specimens (51%). Additionally, 90% of histology reports for TBLN cases were consistent with TB.

### 3.2. Validity and Comparative Performance of Diagnostic Tests

Table 4 displays the diagnostic validity of various tests using two different reference standards: culture and CRS. The performance metrics include sensitivity, specificity, positive predictive value (PPV), negative predictive value (NPV), and accuracy. The MPT64 test exhibited the highest sensitivity with both reference standards, particularly for TBLN (91% with culture and 84% with CRS) and TBP (57% with culture and 51% with CRS). Culture exhibited moderate sensitivity (59% for TBLN and 31% for TBP) when using CRS as the reference. Xpert showed lower sensitivity compared to MPT64 and culture, particularly for TBP. When using CRS as the reference, the sensitivity of Xpert for TBLN was 48% and 7% for TBP, whereas when using culture as the reference, it was 61% for TBLN and 10% for TBP.

The culture and acid-fast stain demonstrated the highest specificity (100%) across all cases and reference standards. Xpert showed high specificity: 74% for TBLN and 95% for TBP when using culture as the reference, and 88% for TBLN and 100% for TBP when using CRS as the reference. The MPT64 test exhibited the lowest specificity among the evaluated tests. The test showed a specificity of 37% when using culture as the reference and 15% when using CRS as the reference for TBLN. The specificity for TBP was notably low, showing 0% with CRS as the reference and 49% with culture as the reference.

Figure 3 illustrates the overlap between the MPT64, culture, and Xpert results. Among the MPT64-positive cases, 122 (51%) also had positive cultures, and 90 (37%) were Xpert-positive, suggesting concordance. Notably, the MPT64 test identified 151 (63%) of the Xpert-negative cases and 119 (49%) of the culture-negative EPTB cases as positive, indicating its potential utility in detecting EPTB cases that other tests may miss.

Table 5 outlines the diagnostic performance of the MPT64 test, culture, Xpert, and AFB staining with various clinical and histopathological parameters, including histopathological groups (HPGs), diagnostic delays, and disease severity. The MPT64 antigen test showed high sensitivity (85%) across all HPGs suggestive of TB (STB). Specifically, it demonstrated 100% positivity in HPG-1 (well-formed granulomas without necrosis) and HPG-4 (necrosis only). Acid-fast staining, on the other hand, displayed very low sensitivity across all HPGs, highlighting its limited diagnostic utility, particularly in cases with complex histopathology.

### 3.3. Impact of Diagnostic Delays and Disease Severity on Test Performance

Diagnostic delays had a noticeable impact on the positivity rates of the tests, particularly for culture and Xpert. In TBLN cases with delayed diagnosis, MPT64 maintained a moderate positivity rate of 85%, whereas culture and Xpert had lower positivity rates of 57% and 45%, respectively. In contrast, when diagnosis was not delayed, MPT64 positivity slightly decreased to 83%, with culture and Xpert showing slight improvements to 59% and 53%. A similar trend was observed in TBP cases, where delayed diagnosis significantly reduced the sensitivity of culture, Xpert, and MPT64, underscoring the critical importance of timely diagnostic intervention.

Disease severity influenced diagnostic test performance. In TBLN patients with severe disease, MPT64 showed a positivity rate of 82%, while culture and Xpert exhibited moderate positivity rates (67% and 40%, respectively). In patients with less severe disease, the MPT64 and Xpert tests showed slightly improved positivity, while the culture test showed decreased positivity from 67% to 49%. In TBP patients, severe disease was associated with a substantial drop in the sensitivity of MPT64, culture, and Xpert.

Appendix A presents a comparison of the antigen load (based on staining intensity) across various clinical features. Mild to moderate antigen loads were more frequently observed compared to higher loads. No statistically significant association was found between antigen load and various histopathological features, severity of disease, or delay in diagnosis.

## 4. Discussion

This study evaluated the performance of the MPT64 test in diagnosing EPTB, specifically TBLN and TBP, in a high-TB, low-HIV, resource-limited setting. Using both culture and CRS, the MPT64 test demonstrated higher overall sensitivity (81% (95% CI: 74–87) with culture and 70% (95% CI: 65–75) with CRS) compared to other diagnostic tests, particularly for TBLN cases. In contrast, Xpert MTB/RIF showed an overall sensitivity of 47% with CRS and 44% with culture.

For TBLN, the MPT64 test demonstrated a higher sensitivity of 91% (CI: 84–95) using culture as the reference, compared to 61% (CI: 51–70) for Xpert. For TBP, MPT64 showed a sensitivity of 57% (CI: 41–72), while Xpert showed a sensitivity of 10% (CI: 3–27). Using the composite reference standard (CRS), MPT64 achieved 84% sensitivity (CI: 78–89) for TBLN and 51% (CI: 43–60) for TBP, whereas Xpert showed 48% (CI: 40–55) for TBLN and 7% (CI: 3–12) for TBP. This difference underscores the impact of the disease site on the performance of diagnostic tests and highlights the MPT64 test’s particular strength in diagnosing TBLN over TBP. A previous study using CRS as the reference also showed the higher sensitivity of the MPT64 test for TBLN compared to TBP [15].

The head-to-head comparisons revealed substantial overlap between the tests: 51% of culture-positive and 37% of Xpert-positive cases were also MPT64-positive. Notably, MPT64 detected TB in 63% of Xpert-negative cases and 49% of culture-negative cases, suggesting its additional diagnostic value in identifying EPTB cases that may be missed by other methods. These findings underscore the potential utility of MPT64 in enhancing EPTB detection and highlight the benefit of combining multiple diagnostic approaches for more accurate diagnosis.

The MPT64 test consistently showed higher sensitivity across patient groups and reference standards, with an overall sensitivity of 81% compared to 47% for Xpert when using culture as the reference. This trend was more pronounced in TBLN cases, where MPT64 sensitivity reached 91% and Xpert reached 61%. For TBP, both tests exhibited lower sensitivity, but MPT64 still surpassed Xpert, with 57% versus 10% sensitivity, respectively. While Xpert maintained higher specificity (85% overall) compared to MPT64 (37%), the added sensitivity of MPT64 is especially valuable in resource-limited settings, where missed diagnoses can have serious consequences. One possible explanation for the better performance of the MPT64 test in TBP patients might be the use of the cell block preparation from pleural fluid aspirated samples. Cell blocks preserve cellular architecture and increase cellularity, allowing for a more detailed observation of pathological features. However, this study did not directly compare outcomes between cell block and non-cell block samples, nor did it stratify by sample preparation method. Additionally, cell blocks enable the preparation of multiple sections from a single sample, facilitating comprehensive evaluations with various staining techniques and repeated examinations. These strengths contribute to higher diagnostic accuracy and reliability, making cell blocks a superior method for diagnosing EPTB compared to traditional acid-fast staining [27]. The significantly higher sensitivity of the MPT64 test (81% CI: 74–87, using culture as the reference; and 70% CI: 65–75, using CRS as the reference) on cell blocks in our study provides a distinct diagnostic advantage over the smear-based methods used in Zanzibar and Tanzania [15,28]. This enhanced sample quality allows for better antigen detection, as reflected in the higher sensitivity of the MPT64 test. While both the Zanzibar and Tanzania studies demonstrated the potential of MPT64 in improving EPTB diagnosis in resource-limited settings, the sensitivity was not as high as in our study. This suggests that the performance of the MPT64 test can be significantly improved when paired with better sampling methods like cell blocks [15,28].

In settings with limited diagnostic infrastructure, a test with high sensitivity is critical to avoid missed cases and reduce disease progression. The MPT64 test’s higher sensitivity makes it a valuable tool for identifying TB cases that other tests might miss, as missed diagnoses can lead to disease deterioration. The higher sensitivity of the MPT64 test compared to the Xpert test can be attributed to fundamental differences in their detection mechanisms: antigen accumulation versus DNA detection. Antigens like MPT64 seem to accumulate in inflammatory cells even when the bacterial concentration is low. This accumulation allows the MPT64 test to identify TB cases with greater sensitivity, particularly in scenarios where bacterial DNA might not be as readily accessible or present in sufficient quantities for detection by nucleic acid tests like Xpert. If the bacilli are sparse, as can happen in EPTB cases, the Xpert test may fail to detect the presence of TB, leading to lower sensitivity.

The culture test had lower sensitivity for TBP (31%) and TBLN (59%), aligning with the challenge of growing mycobacteria from pleural fluid, which might have fewer bacilli and be less conducive to culture growth compared to lymph node specimens [29]. The higher culture positivity observed in TBLN compared to TBP may be attributed to a greater bacterial load, resulting in increased sensitivity [6].

The specificity of the MPT64 test was low. However, the limited number of non-TB cases makes it difficult to make a firm assessment of specificity. One explanation could be the possibility of concomitant comorbidities with TB, such as cancers and other bacterial conditions, leading to the categorization of patients as non-TB, despite underlying TB. This study was designed to assess the performance of the test in routine diagnostic settings, and patients presumed to have TB were included. A control group with known non-TB cases was not included. The culture test had perfect specificity (100%), confirming TB without false positives. The culture test’s perfect specificity emphasizes its role as the gold standard for TB diagnosis. However, its time-consuming nature, as well as the need for highly sophisticated infrastructure, equipment, and skilled human resources, in addition to its high cost, limits its use as a robust diagnostic tool, particularly in a resource-constrained setting [30]. The Xpert test also showed high specificity of 100% for TBP and 88% for TBLN. However, the lower sensitivity for TBLN must be considered in clinical decision-making. Clinicians should consider a comprehensive approach integrating the MPT64 test with other diagnostic tests, particularly Xpert, and taking clinical context into account for better EPTB diagnosis and management outcomes. Despite its lower specificity, the MPT64 test’s ability to detect a greater number of true TB cases outweighs the drawbacks of potential false positives, especially in settings where the consequences of missing a TB case are severe. The high sensitivity of MPT64 is particularly advantageous for timely diagnoses and can be effective for TB control.

In this study, we also evaluated the MPT64 test according to the morphological features of lymph node biopsies, finding that 85% of the reported cases showing features consistent with TB were identified as positive by the MPT64 test. The high concordance between the morphological patterns and positive MPT64 test results reinforces the test’s potential as a reliable diagnostic tool for EPTB, particularly for TBLN.

Investing in new diagnostic tests, such as MPT64, which is a simple, rapid, robust, and sensitive test, and making incremental improvements in laboratory infrastructure can provide valuable support for the diagnosis of EPTB and enhance diagnostic capabilities over time in high-burden-TB settings. Any clinical pathology laboratory can easily obtain and process even tiny specimens for MPT64 testing, which is another significant benefit [20,31]. Therefore, the MPT64 test should be a diagnostic procedure in the routine diagnostic protocol for EPTB cases.

There are also some limitations to our study, including that the clinically presumed EPTB patients and response to ATT may not offer a correct diagnosis of TB; hence, the CRS used may have less specificity. Additionally, the low specificity of MPT64 was a major limitation. Only 18 patients were ultimately categorized as non-TB, and even among these, misclassification was possible due to comorbidities. This is due to the study design that assessed the MPT64 test in a routine programmatic setup for TB control, ensuring it did not interfere with other routine diagnostic procedures. A properly designed case–control study with a robust number of microbiologically confirmed non-TB cases would better define the test’s specificity. Furthermore, the lack of uniform availability of radiological imaging (including chest X-rays and CT scans) across all cases precluded a standardized assessment of disease extent. This parameter could have provided important complementary context to the histopathological features and immunostaining results. Future studies should aim to include such a cohort to clarify the potential for false positives, especially in patients with granulomatous diseases, malignancies, or inflammatory conditions. Finally, the study setting, being a single tertiary-care hospital, may limit generalizability.

## 5. Conclusions

The MPT64 test shows promising sensitivity for diagnosing EPTB, particularly TBLN, in high-burden, resource-limited settings. Its integration into routine diagnostics as a complementary tool alongside Xpert may enhance TB case detection, especially where culture is not feasible. However, further prospective studies with more rigorous diagnostic standards, larger control groups, and stratified sampling methods are essential to validate its diagnostic utility and better define its role in routine clinical algorithms.

## Figures and Tables

**Figure 1 pathogens-14-00741-f001:**
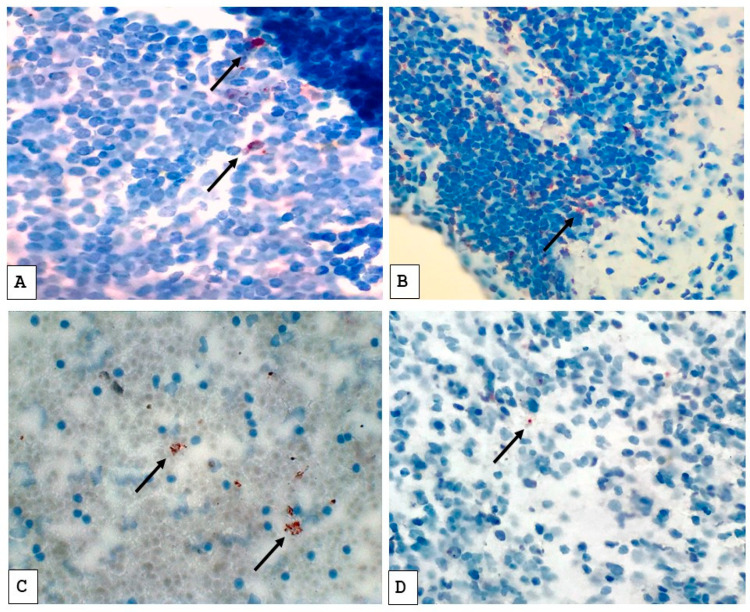
Immunostaining pattern with anti-MPT64 antibody in tuberculous lymph node biopsies (**A**,**B**) and cell blocks prepared from tuberculous pleural fluids (**C**,**D**). The positive MPT64 antigen staining is seen as reddish-brown granular deposits in the cytoplasm of cells (as indicated by arrows). (**A**,**C**) Represent stronger staining; (**C**,**D**) represent weaker staining.

**Figure 2 pathogens-14-00741-f002:**
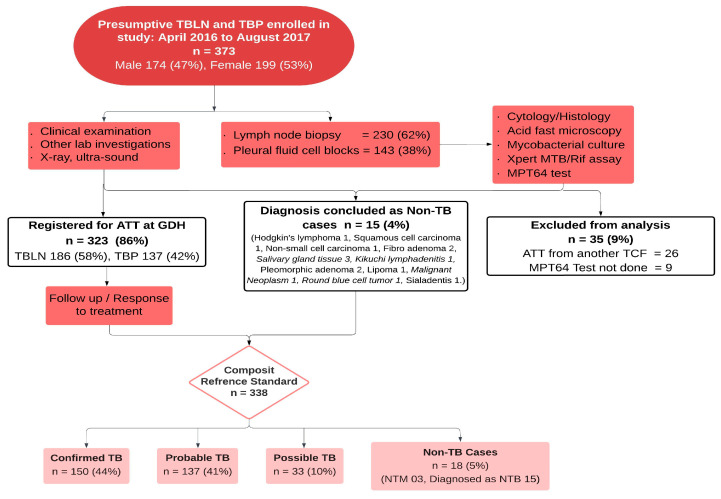
Patient flow and diagnostic categorization of presumptive TBLN and TBP cases using the composite reference standard. Abbreviations: TBLN, tuberculosis lymphadenitis; TBP, tuberculous pleuritis; ATT, anti-tuberculous treatment; GDH, Gulab Devi Hospital; LN, lymph node; TCF, TB care facility; TB, tuberculosis; NTM, non-tuberculous mycobacterium; NTB, non-tuberculous.

**Figure 3 pathogens-14-00741-f003:**
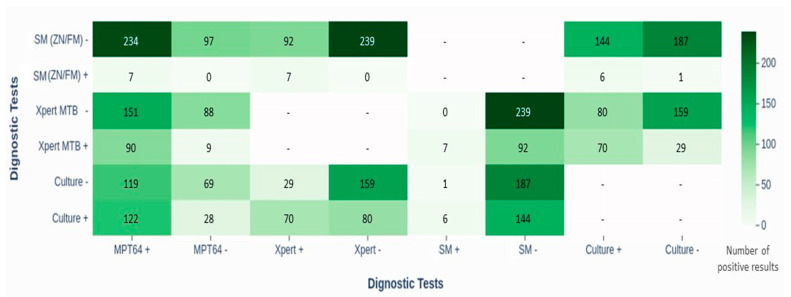
Heat map showing head-to-head comparison of diagnostic tests (MPT64 with Xpert MTB/RIF, smear microscopy (ZN/FM), and culture) in diagnosing TBLN and TBP. The figure highlights overlap in positive results, especially with Xpert and culture, and illustrates the limited sensitivity of smear methods. Abbreviations: SM, smear microscopy; ZN, Ziehl–Neelsen; FM, fluorescence microscopy; Xpert MTB, Xpert *Mycobacterium tuberculosis* assay; LJ, Lowenstein–Jensen; MPT64, immunochemistry-based antigen detection test; TBLN, tuberculosis lymphadenitis; TBP, tuberculous pleuritis.

**Table 1 pathogens-14-00741-t001:** Criteria for classifying patients into different categories under the composite reference standard.

Confirmed TB Case	Positive Mycobacterial Culture
Probable TB case	Clinically presumptive extrapulmonary tuberculosis AND (a) TB lymphadenitis: Histology consistent with TB AND a good response to ATT at 2–3 months and/or end of treatment. (b) TB pleural effusion: Radiological evidence of pleural effusion AND a good response to ATT at 2–3 months and/or end of treatment.
Possible TB case	Clinically presumptive extrapulmonary tuberculosis AND one of the following: A good response to ATT at 2–3 months and/or the end of treatment. TB lymphadenitis: Biopsy—morphological features suggestive of TB AND histology consistent with TB. TB pleural effusion: Radiological evidence of pleural effusion AND pleural fluid protein level ≥ 3 g/dL AND lymphocytosis.
Non-TB case (control subject)	Negative *Mycobacterium tuberculosis* complex and/or detected NTM on culture AND cytology/histology examination concluded specific diagnosis other than TB.
TB, tuberculosis; NTM, non-tuberculous mycobacteria; ATT, anti-tuberculosis treatment. Definitions: Presumptive extrapulmonary tuberculosis case: Symptoms indicative of TB. Good response to ATT: Fulfillment of at least two of three criteria: (1) improvement of presenting symptoms, (2) regression of local disease signs (e.g., pleural effusion, enlarged lymph nodes), and (3) weight gain. Morphologic features consistent with TB: epithelioid granuloma and Langerhans-type giant cell necrosis.	

**Table 2 pathogens-14-00741-t002:** Sociodemographic and clinical characteristics of the study participants.

Characteristics	N = 338	EPTB Cases	Non-EPTB Cases
		n = 320 (95%)	n = 18 (5%)
Gender			
Male	160 (47)	155 (48)	5 (28)
Female	178 (53)	165 (52)	13 (72)
Age (years)			
0–14	54 (16)	53 (17)	1 (6)
15–44	245 (72)	231 (72)	14 (78)
≥45	39 (12)	36 (11)	3 (17)
Presumptive EPTB site			
Lymphadenitis	201 (59)	186 (58)	15 (83)
Pleuritis	137 (41)	134 (42)	3 (17)
HIV status			
Positive	0 (0)	0 (0)	0 (0)
Non-reactive	80 (24)	80 (25)	0 (0)
Not done	258 (76)	240 (75)	18 (100)

EPTB, extrapulmonary tuberculosis; HIV, human immunodeficiency virus.

**Table 3 pathogens-14-00741-t003:** Diagnostic test results showing positive results across different methods.

Characteristics	N	Positive Test Results/Total Tests (%)			
		EPTB Cases		Non-EPTB Cases	
		TBP	TBLN	TBP	TBLN
TB diagnostic tests					
Microscopy (ZN/FM)	338	0/134 (0)	7/186 (4)	0/3 (0)	0/15 (0)
Xpert MTB/Rif assay	338	9/134 (7)	89/186 (48)	0/3 (0)	1/15 (7)
Culture	338	42/134 (31)	108/186 (58)	0/3 (0)	0/15 (0)
MPT64	338	69/134 (51)	157/186 (84)	3/3 (100)	12/15 (80)
Histopathology reports					
Consistent with TB	168	0/134 (0)	168/186 (90)	0/3 (0)	0/15 (0)
Concluded as non-TB	15	0/134 (0)	0/186 (0)	0/3 (0)	15/15 (100)
Others	17	0/134 (0)	17/186 (9)	0/3 (0)	0/15 (0)
Not done	138	134/134 (100)	1/186 (1)	3/3 (0)	0/15 (0)

EPTB, extrapulmonary tuberculosis; TB, tuberculosis; ZN/FM, Ziehl–Neelsen and fluorescence microscopy; TBP, tuberculous pleuritis; TBLN, tuberculous lymphadenitis; MTB/Rif assay, *Mycobacterium tuberculosis* rifampicin assay; MPT64, immunochemistry-based antigen detection test.

**Table 4 pathogens-14-00741-t004:** Diagnostic performance of MPT64, Xpert MTB/RIF, and smear microscopy against culture and CRS.

Characteristics	Sensitivity	Specificity	PPV	NPV	Accuracy
	% (95% CI)	% (95% CI)	%	%	%
Culture as reference					
All cases					
MPT64 test	81 (74.1–87.2)	37 (29.8–44.0)	51	71	56
Xpert assay	47 (38.5–55.0)	85 (78.6–89.4)	78	66	68
SM (ZN/FM)	4 (1.5–8.5)	99 (97.1–99.9)	86	56	57
TB lymphadenitis					
MPT64 test	91 (83.6–95.5)	24 (15.5–33.6)	58	69	60
Xpert assay	61 (51.2–70.3)	74 (64.1–82.1)	73	62	67
SSM (ZN/FM)	6 (2.1–11.7)	99 (94.2–99.9)	86	47	49
TB pleuritis					
MPT64 test	57 (41.0–72.3)	49 (39.1–59.9)	33	72	52
Xpert assay	10 (2.7–22.6)	95 (88.1–98.3)	44	70	69
SM (ZN/FM)	0 (0.0–8.4)	100 (96.1–100)	-	69	69
CRS as reference					
All cases					
MPT64 test	70 (65.0–75.4)	15 (3.2–37.9)	93	3	67
Culture	47 (41.2–52.8)	100 (83.1–52.8)	100	11	50
Xpert assay	31 (25.5–35.9)	90 (68.3–98.8)	98	8	34
SM (ZN/FM)	2 (0.8–4.5)	100 (83.1–100)	100	6	8
TB lymphadenitis					
MPT64 test	84 (78.2–89.2)	18 (3.8–43.4)	92	9	79
Culture	59 (51.2–65.9)	100 (80.5–100)	100	18	62
Xpert assay	48 (40.4–55.3)	88 (63.6–98.5)	98	13	51
SM (ZN/FM)	4 (1.5–7.7)	100 (80.5–100)	100	9	12
TB pleuritis					
MPT64 test	51 (42.7–60.2)	0 (0.0–70.8)	96	0	50
Culture	31 (23.6–39.9)	100 (29.2–100)	100	3	33
Xpert assay	7 (3.1–12.4)	100 (29.2–100)	100	2	9
SM (ZN/FM)	0 (0.0–2.7)	100 (29.2–100)	-	2	2

PPV, positive predictive value; NPV, negative predictive value; CI, confidence interval; CRS, composite reference standard; SM, smear microscopy; ZN, Ziehl–Neelsen staining; FM, fluorescence microscopy; TB, tuberculosis; TBLN, tuberculous lymphadenitis; TBP, tuberculous pleuritis.

**Table 5 pathogens-14-00741-t005:** Association of histopathology, diagnostic delays, and disease severity with diagnostic test results.

	n	MPT64		Culture		Xpert		SM (ZN/FM)	
		+	−	+	−	+	−	+	−
		n (%)	n (%)	n (%)	n (%)	n (%)	n (%)	n (%)	n (%)
Morphology STB	168								
HPG-1	3	3 (100)	0 (0)	2 (67)	1 (33)	1 (33)	2 (67)	0 (0)	3 (100)
HPG-2	90	73 (81)	17 (19)	58 (64)	32 (36) *^1^	39 (43)	51 (57) ^^3^	2 (2)	88 (98)
HPG-3	65	57 (88)	8 (12)	35 (54)	30 (46) *^1^	34 (52)	31 (48)	3 (5)	62 (95)
HPG-4	10	10 (100)	0 (0)	7 (70)	3 (30)	7 (70)	3 (30)	1 (10)	9 (90)
Morphology NSTB	32								
HPG-0	17	13 (76)	4 (24)	6 (35)	11 (65)	7 (41)	10 (59) ^^1^	1 (6)	16 (94)
HPG-5	15	12 (80)	3 (20)	0 (0)	15 (10) *^1^	1 (7)	14 (93) ^^1^	0 (0)	15 (100)
Diagnostic delay TBLN	186								
Delayed	110	94 (85)	16 (15)	63 (57)	47 (43) *^1^	49 (45)	61 (55) ^^3^	5 (5)	105 (98)
Not-delayed	76	63 (83)	13 (17)	45 (59)	31 (41) *^1^	40 (53)	36 (43) ^^1^	2 (3)	74 (97)
Diagnostic delay TBP	134								
Delayed	51	27 (53)	24 (47)	17 (33)	34 (67)	4 (8)	47 (92) ^^2^	0 (0)	83 (100)
Not-delayed	83	42 (51)	41 (49)	25 (30)	58 (70)	5 (6)	78 (94) ^^4^	0 (0)	51 (100)
Disease severity TBLN	201								
Severe	57	47 (82)	10 (18)	38 (67)	19 (33)	23 (40)	34 (60) ^^3^	5 (9)	52 (91)
Less severe	144	122 (85)	22 (15)	70 (49)	74 (51) *^3^	67 (47)	77 (53) ^^2^	2 (1)	142 (99)
Disease severity TBP	137								
Severe	58	22 (38)	36 (62)	19 (33)	39 (67)	5 (6)	74 (94) ^^1^	0 (0)	58 (100)
Less severe	79	50 (63)	29 (37)	23 (29)	56 (71) *^3^	4 (7)	54 (93) ^^5^	0 (0)	79 (100)

SM, smear microscopy; ZN, Ziehl–Neelsen; FM, fluorescence microscopy; LJ: Lowenstein–Jensen; HPG, histopathological groups; STB, suggestive of tuberculosis; NSTB, non-suggestive of tuberculosis; TBLN, tuberculous lymphadenitis; TBP, tuberculous pleuritis. * Superscript numbers represent NTM cases. ^ Superscript numbers represent error results.

## Data Availability

The datasets generated during and/or analyzed during the current study, “Advancing extrapulmonary tuberculosis diagnosis: potential of MPT64 Immunochemistry-based antigen detection test in a high TB, low HIV endemic setting,” are available from the corresponding author on reasonable request. The author may be contacted at the Centre for International Health, Department of Global Public Health and Primary Care, University of Bergen, PB 7804, 5020 Bergen, Norway, or via email: a.wali@uib.no and/or dr_wali786@yahoo.com.

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
