# Peer review of "Advancing Extrapulmonary Tuberculosis Diagnosis: Potential of MPT64 Immunochemistry-Based Antigen Detection Test in a High-TB, Low-HIV Endemic Setting"

_pathogens, 2025, doi:10.3390/pathogens14080741_

Round 1

Reviewer 1 Report

Comments and Suggestions for Authors

Wali and colleagues evaluated the performance of a test based on MPT64 antigen detection (immunostaining) for the diagnosis of extrapulmonary tuberculosis. Overall, the manuscript is technically sound. However, the authors should review some issues:

As the authors mention that it is a new technique, more details about the test should be added in the methodology. Only the interpretation of the results was presented in the manuscript. Reference 20, presented on line 89, did not use the methodology based on MPT64-immunodetection.

According to the authors “The study population was patients of all ages presenting with presumptive EPTB, prospectively enrolled from the outpatients’ department between April 2016 and August 2017.” Were the patients still being treated for tuberculosis and recruited for the study, or was the material collected during the enrollment period? Please clarify.

Table titles and figure legends should be revised. They are describing how the results have been presented. Also, there is no need to state that it is a table or a figure at the beginning of the titles.

Lines 341: the reference is missing

Line 356: please remove “or degraded”. As Gene Xpert is based on nucleic acid amplification, the bacteria being “degraded” may not affect detection.

Please add italics for Mycobacterium tuberculosis.

There are minor typographical errors that need to be revised as well.

Author Response

Comment 1: As the authors mention that it is a new technique, more details about the test should be added in the methodology. Only the interpretation of the results was presented in the manuscript. Reference 20, presented on line 89, did not use the methodology based on MPT64-immunodetection.

Response 1: We appreciate this important observation. In the revised manuscript, we have added a brief description of the MPT64 antigen detection in the “Assessment of immunostaining” section of the Methods. The revised text now reads:

“ICC and IHC for MPT64 were performed using an in-house polyclonal anti-MPT64 antibody (1:250 dilution), with detection via the Dako Envision+ System-HRP kit (K4009, Dako, Denmark). For ICC, smears were rehydrated, endogenous peroxidase activity was blocked, and slides were incubated with the primary antibody for 60 minutes, followed by HRP-conjugated polymer detection for 45 minutes. Visualization was achieved using 3-amino-9-ethylcarbazole (AEC) substrate and counter-staining with Mayer’s hematoxylin. For IHC, formalin-fixed paraffin-embedded sections underwent deparaffinization and microwave-based antigen retrieval in citrate buffer (pH 6.2) before following the same staining protocol as ICC. Slides were mounted using Immu-Mount (Thermo Fisher Scientific), and wash steps were carried out using Dako Wash Buffer (S3006) [15, 23, 24].”

Additionally, we have corrected the reference on line 89, as the previous citation did not represent the technique used in our study. More relevant references, numbers 12, 18, 20, and 21, have been included to support the methodology. The changes in the revised manuscript are on lines 127 to 136.

Comment 2: According to the authors, “The study population was patients of all ages presenting with presumptive EPTB, prospectively enrolled from the outpatients’ department between April 2016 and August 2017.” Were the patients still being treated for tuberculosis and recruited for the study, or was the material collected during the enrollment period? Please clarify.

Response 2: Thank you for pointing out the ambiguity. We have clarified the statement in the Study Population subsection. The revised text now reads:

“Patients of all ages presenting with clinical suspicion of EPTB were prospectively enrolled from the outpatient department between April 2016 and August 2017. Clinical specimens were collected during the initial presentation/enrollment period, prior to the initiation of anti-tuberculosis treatment.”

This clarification is now reflected in the revised manuscript, and the changes are on lines 93 to 97.

Comment 3: Table titles and figure legends should be revised. They are describing how the results have been presented. Also, there is no need to state that it is a table or a figure at the beginning of the titles.

Response 3: We thank the reviewer for this suggestion. All table titles and figure legends have been revised to enhance clarity and briefness. Redundant phrases such as “Table showing…” or “Figure displaying…” have been removed, and the titles now directly describe the data or findings presented.

Comment 4: Line 341: the reference is missing.

Response 4: We apologize for the oversight. The missing reference has now been added at positions 15 and 28 in the reference list and cited correctly in line 316.

Comment 5: Line 356: Please remove “or degraded.” As GeneXpert is based on nucleic acid amplification, the bacteria being “degraded” may not affect detection.

Response 5: We agree with the reviewer’s observation. The phrase “or degraded” has been removed from line 359 to accurately reflect the capabilities of the GeneXpert.

Comment 6: Please add italics for Mycobacterium tuberculosis.

Response 6: We thank the reviewer for pointing this out. All instances of Mycobacterium tuberculosis in the manuscript have been italicized in accordance with scientific nomenclature conventions.

Comment 7: There are minor typographical errors that need to be revised as well.

Response 7: A comprehensive review of the manuscript has been conducted to correct all typographical errors. We have ensured proper grammar, punctuation, and spelling throughout the revised document.

Reviewer 2 Report

Comments and Suggestions for Authors

The Authors present a comparative study between the MPT64 antigen detection test and the reference methods for the diagnosis of EPTB. The purpose of the study is of high importance, but it suffers from the lack of a control group to evaluate the method's specificity. The data obtained on the small group of non-TB cases indicates a very low specificity. Although other methods guarantee a specificity close to 100%, it is implausible to think that a candidate method for diagnostic purposes should be supported by another method. It is therefore recommended to extend the analyses to a control group to complete the validation.
Specific comments:
- lines 31-32: the manuscript still tests and reports the specificity of the MPT64 method and compares it with the reference methods; therefore, for transparency, the specificity value obtained from the evaluation should be reported;
- line 148: "This is table" is a typo that should be removed. The same applies to the other tables;
- Figure 2: The figure resolution and text size should be increased;
- Table 5: Which correlation analysis test was applied? Are frequencies reported in the table; how can they indicate correlations?
- Table S1: Supplementary data should not be reported in the main text.

Author Response

Comment 1: The authors present a comparative study between the MPT64 antigen detection test and the reference methods for the diagnosis of EPTB. The purpose of the study is of high importance, but it suffers from the lack of a control group to evaluate the method's specificity. The data obtained on the small group of non-TB cases indicates a very low specificity. Although other methods guarantee a specificity close to 100%, it is implausible to think that a candidate method for diagnostic purposes should be supported by another method. It is therefore recommended to extend the analyses to a control group to complete the validation.

Response 1: We appreciate the reviewer’s observation regarding the need for an adequate control group. The current study was primarily designed as a diagnostic performance evaluation in presumptive EPTB cases in routine care with the assumption that some presumptive cases will be non-TB and will serve as controls. However, unlike our experience from other sites of implementation of this test, in Pakistan the non-TB cases were few. We acknowledge the limitation of having a small number of confirmed non-TB cases, which affected the specificity estimate. We have clearly stated this limitation in the Discussion section and emphasized the need for a future, larger study including a properly matched control group to validate and refine the specificity of the MPT64 antigen detection method. Due to resource constraints and the nature of our prospective enrollment, we were unable to incorporate a healthy or disease control group in the present analysis.

This clarification is reflected in the revised manuscript on lines 373 to 377.

Comment 2: Lines 31–32: the manuscript still tests and reports the specificity of the MPT64 method and compares it with the reference methods; therefore, for transparency, the specificity value obtained from the evaluation should be reported.

Response 2: Thank you for this important observation. The specificity value obtained from our evaluation is indeed reported in the Results section on lines 223–230 in the revised manuscript. However, due to word count limitations in the abstract, we were unable to include this detail there. We agree that transparency is important and are open to incorporating the specificity value in the abstract if the journal allows a slight extension of the word limit.

Comment 3: Line 148: "This is table" is a typo that should be removed. The same applies to the other tables.

Response 3: We apologize for the typographical error. All instances of “This is table” and similar phrases have been corrected or removed from the manuscript and table legends to ensure clarity and consistency.

Comment 4: Figure 2: The figure resolution and text size should be increased.

Response 4: We appreciate this suggestion. Figure 2 has been updated with improved resolution and larger, legible text to enhance readability in both digital and print formats. The revised figure has been included in the revised manuscript file.

Comment 5: Table 5: Which correlation analysis test was applied? Are frequencies reported in the table; how can they indicate correlations?

Response 5: Thank you for highlighting this issue. We acknowledge the confusion. Table 5 was intended to show associations rather than statistical correlation. We have now updated the table title and legend to reflect that a chi-square test was applied to assess the association between MPT64 results and reference method outcomes.

Comment 6: Table S1: Supplementary data should not be reported in the main text.

Response 6: We agree with the reviewer. Table S1 has now been removed from the main manuscript and correctly placed in the supplementary materials section. A brief reference to the supplementary table is included in the main text to guide readers appropriately.

This reference is reflected in the revised manuscript on lines 391 to 393.

Reviewer 3 Report

Comments and Suggestions for Authors

This article estimates the role of a rapid method as MPT64 to be found as a reliable and cheaper method for confirming MTB. Novel technologies for rapid identification of the culture isolates and Anti tubercular drug resistant isolates should become a top priority and MPT s first bibliography comes since 2011. Although, not for direct detection in sputum or blood, it seems that works perfectly in low-resource settings which do not require molecular testing.

The content of the text is based on targeted bibliographic references and the discussion contains the most important studies. All tables are easy to read and understand and concise. It is well written with no detection detected.

However, the authors should comment on some parameters:

  • The possibility of malnutrition among participants that may affect the course of Tb disease
  • The existence of major comorbitidies such as cancer, renal disease and/or immunosuppresion
  • The co existence of pulmonary and extrapulmonary disease
  • The extent of the disease in radiological and CT images
  • Prior therapy with antiTB drugs that may alter the test

Author Response

Reviewer Comment 1: The possibility of malnutrition among participants that may affect the course of TB disease.

Response comment 1: Thank you for highlighting this important aspect. We acknowledge that malnutrition is a significant risk factor for both the development and progression of TB. However, nutritional status (e.g., BMI, serum albumin, or other markers) was not systematically documented in our study cohort and therefore could not be incorporated into the current analysis. We will consider including nutritional assessments in future our studies to better interpret the performance of immunohistochemical diagnostics in TB, particularly in resource-constrained settings where malnutrition is prevalent.

Reviewer Comment 2: The existence of major comorbidities such as cancer, renal disease and/or immunosuppression.

Response comment 2: We agree that comorbidities such as malignancy, chronic kidney disease, and immunosuppressive states (including HIV) can influence both TB manifestations and the host immune response, potentially affecting the sensitivity of immunohistochemical methods. In our study, patients with known HIV infection or those undergoing active immunosuppressive therapy were excluded. However, detailed screening for other comorbidities such as cancer or renal disease was not part of the study protocol. We have now clarified these inclusion criteria in the "Methods" section (lines 101–102) of the revised manuscript.

Reviewer Comment 3: The co-existence of pulmonary and extrapulmonary disease.

Response comment 3: We appreciate this observation. Our study focused specifically on extrapulmonary TB involving lymph nodes and pleural tissues, clinically presumed. While some participants may have had concurrent pulmonary TB, our inclusion criteria were based solely on tissue biopsies from extrapulmonary sites. Unfortunately, comprehensive imaging or microbiological confirmation of concurrent pulmonary involvement was not consistently available and therefore not analyzed. A clarifying note has been added to the “Study Population” subsection (lines 97-98).

Reviewer Comment 4: The extent of the disease in radiological and CT images.

Response comment 4: Radiological imaging (including chest X-rays and CT scans) was not uniformly available across all cases. As a result, disease extent based on imaging could not be standardized within our cohort. We agree that this is a valuable parameter that could have provided additional context to histopathological and immunostaining findings. This limitation has now been acknowledged in the "Limitations" section (lines 382–385), with recommendations for its inclusion in future studies.

Reviewer Comment 5: Prior therapy with anti-TB drugs that may alter the test.

Response comment 5: This is an important consideration. To minimize confounding, we included only newly diagnosed patients with no prior history of anti-tuberculous treatment before biopsy and tissue sampling. This criterion is explicitly stated under the "Inclusion Criteria" in sub-section 2.1. “Study Setting and Population” of the Methods section (lines 96–97).

Round 2

Reviewer 2 Report

Comments and Suggestions for Authors

Although the limited number of controls remains a critical issue, the authors have highlighted this limitation of the study in the manuscript. They have also addressed the other comments. The results obtained nevertheless have strong translational impact, and the manuscript can be published in its current form.